# COVID-19 Vaccine Mandates and Vaccine Hesitancy among Black People in Canada

**DOI:** 10.3390/ijerph20237119

**Published:** 2023-11-28

**Authors:** Aisha Giwa, Morolake Adeagbo, Shirley Anne Tate, Mia Tulli-Shah, Bukola Salami

**Affiliations:** 1College of Health Sciences, School of Public Health University of Alberta, Edmonton, AB T6G 1C9, Canadatulli@ualberta.ca (M.T.-S.); oluwabukola.salami@ucalgary.ca (B.S.); 2Department of Sociology, University of Alberta, Edmonton, AB T6G 2R3, Canada; shirleya@ualberta.ca

**Keywords:** COVID-19, vaccine mandates, Black Canadians, biopower, governmentality, vaccine hesitancy

## Abstract

Objectives: COVID-19 vaccine mandates increased vaccination rates globally. Implemented as a one-size-fits-all policy, these mandates have unintended harmful consequences for many, including Black Canadians. This article reports findings on the interconnectedness of vaccine mandates and vaccine hesitancy by describing a range of responses to mandatory COVID-19 vaccination policies among Black people in Canada. Methods: Using qualitative research methods, semi-structured interviews with 36 Black people living in Canada aged 18 years and over across 6 provinces in Canada were conducted. Participants were selected across intersectional categories including migration status, income, religion, education, sex, and Black ethnicity. Thematic analysis informed the identification of key themes using Foucauldian notions of biopower and governmentality. Results: Our results show how the power relations present in the ways many Black people actualize vaccine intentions. Two main themes were identified: acceptance of the COVID-19 vaccine in the context of governmentality and resistance to vaccine mandates driven by oppression, mistrust, and religion. Conclusion: COVID-19 vaccine mandates may have reinforced mistrust of the government and decreased confidence in the COVID-19 vaccine. Policy makers need to consider non-discriminatory public health policies and monitor how these policies are implemented over time and across multiple sectors to better understand vaccine hesitancy.

## 1. Introduction

The COVID-19 vaccine is one of the most effective ways of limiting the spread of the virus, reducing the severity of the disease and strain on the economic and healthcare system [1,2,3]. COVID-19 vaccine mandates—policies that limit the participation of individuals in certain activities including employment, travel, and other social amenities without proof of vaccination—were enacted by governments to contain the spread of the virus and increase vaccination rates [2,3]. While mass vaccination against COVID-19 started in December 2020 among priority populations, COVID-19 vaccine mandates were introduced in all jurisdictions across Canada, requiring vaccine certificates to access nonessential services. Additionally, provinces like Alberta and Saskatchewan accepted proof of a negative COVID-19 test on a case-by-case basis [3]. These mandates were expected to increase the uptake of the vaccine and slow the spread of the virus. Research shows that COVID-19 vaccine mandates can increase vaccine uptake particularly just after vaccination mandates and policies were first announced [3,4,5]. COVID-19 vaccine mandates have had a disproportionate impact on certain populations including Black people and other visible minority groups and may have contributed to vaccine hesitancy. For example, research in Canada suggests that due to overrepresentation in the frontline and service sectors in Canada, visible minorities experienced more vulnerability to COVID-19 than White Canadians [6]. Public health measures such as vaccine and mask mandates, lockdowns, travel restrictions, social distancing, etc., increase the vulnerabilities of many populations including visible minorities while limiting movement within and across national and international borders, economic activity, and access to social and other services attempted to contain the spread of the virus [7]. However, confidence in the vaccine was poor, particularly among Black people and other ethnic minorities in the United States and United Kingdom due to a legacy of mistrust in healthcare/science, conspiracy theories spread by social media, concerns about vaccine safety, and misinformation, among other factors [8,9].

Vaccine hesitancy, “defined as a delay in acceptance or refusal of vaccination despite the availability of vaccine services,” is a global health problem [10]. Vaccine hesitancy impacts vaccination rates by reducing vaccine uptake or coverage, and it indirectly impacts vaccination rates by resisting mandatory vaccine policies [10,11]. Vaccine hesitancy among Black people, including those living in Canada, is complex, with a myriad of factors contributing to this phenomenon. Research suggests vaccine hesitancy in this population is caused by misinformation, racism in healthcare, low health literacy, conspiracy theories, mistrust in the government and health industry, religion, concerns about vaccine safety, side effects, and polarizing messages from the government and health authorities [12,13,14,15,16]. Vaccine mandates can fuel mistrust of the government and heighten health, social, and economic inequities [17]. Still, a vaccination campaign with emphasis on mandatory vaccination was launched across provinces and territories to reduce the case counts of COVID-19 throughout Canada and prompted debates about bodily autonomy, infringement of rights and freedoms, and coercion to accept vaccine mandates and the COVID-19 vaccine [18]. Others have argued that vaccine mandates had damaging effects on public trust, vaccine confidence, public health inequities, political polarization, and disruptive protests against public health measures [18,19].

### Theoretical Framework

Understanding vaccine hesitancy among Black people in Canada demands analysis grounded in historical and ongoing systemic and medical racism, including the use of medical technologies for biopolitical control. Thus, this paper draws on Michel Foucault’s (1978) concept of biopower to highlight the ways biological technology has been used in Canada to control Black populations [20,21]. Biopower enacted through vaccine mandates allows the government and other authorities to regulate the health of the population, in this case for the purpose of protecting people from the COVID-19 virus.

Reading vaccinations through a lens of biopolitics offers a way to contextualize COVID-19 vaccines and vaccine mandates as a modern attempt by the Canadian state to assert biopolitical control over populations. For those populations who have historically been targets of biopolitical eugenics and medical experimentation (including Black and Indigenous people in Canada), the COVID-19 vaccine may be viewed as a continuation of this control. Here, we add Charles’ (2023) proposal for nuanced attention to feelings and emotions attached to human behavior. In her work on Vaccines, Hesitancy, and the Affective Politics of Protection in Barbados, Charles (2023) ascribes suspicion to the “gut feelings, emotions, and colonial residues of trauma from biopolitical harms and stimulates forms of skepticism and refusal of potentially lifesaving technologies the human papillomavirus vaccine (HPV) pg. 6” [22]. Like vaccine confidence, suspicion focuses on the quality, safety, and purpose of the vaccines but it also reveals the acceptance of only scientific logic and knowledge-making implications around vaccine refusal as problematic. In other words, the biopolitics of the past can affect vaccine choices of the present and future.

Alongside the concept of biopolitics, we deploy Foucault’s concept of governmentality to understand individuals’ acceptance, hesitancy, and refusal of vaccines. Foucault describes governmentality as the way power is decentered and dispersed through individuals so that there is less need for an overbearing authority to manage the actions of individuals. Instead, individuals and communities become conditioned into self-regulating themselves [23]. Thus, mandates including proof of vaccination engender surveillance and knowledge of who is a good citizen, who acts responsibly, and who acts for the greater good [24]. As theoretical concepts, biopower and governmentality are limited with respect to the way they frame people in general as homogenous objects of control. However, Charles’s work on suspicion emphasizes the ways in which historical and continued trauma from biopolitical harms produces resistance to vaccine mandates [22]. Together, this article frames Black people’s experiences with vaccine mandates as a range of responses that reveal the complexity of vaccine acceptance or refusal.

## 2. Materials and Methods

This qualitative study aimed at understanding the impact of COVID-19 vaccine mandates on vaccine hesitancy among Black people in Canada. We conducted 36 semi-structured interviews with Black people aged 18 years and above across 6 provinces in Canada (Table 1). Purposive sampling was used to recruit participants from an established list of Black Canadians, community leaders, and organizations across Canada. This list was collated and maintained as part of a previous research project on Black people in Canada who consented to participate in future research. Interviews were conducted with participants who responded to our call for interviews and consented to take part in a virtual interview lasting anywhere between 45 and 90 min from February to May 2022. Interviews were conducted with participants of varying income, location, immigration status, and Black ethnicity, among other variables. We obtained signed informed consent from all participants and oral consent from all participants prior to conducting the interviews. Following the interviews, participants received a CAD 25 gift certificate as compensation for their time. Open-ended questions were used to explore (i) issues around perception of risk, (ii) factors that influence COVID-19 vaccine acceptance and vaccine hesitancy, (iii) perceptions towards vaccine mandates, and (iv) strategies to improve vaccine confidence among Black people. These questions were phrased in ways that avoided any assumptions and allowed participants to talk freely. The interviews were conducted by MA. MA took detailed notes during the interview. MA, AG, and BS provided feedback on the results during regular team meetings. All interviews were audio-recorded and transcribed verbatim.

Data analysis was inductive [25]. After transcribing the interviews, a coding framework was developed based on existing research and analysis of a quarter of the interviews (*n* = 9) and reviewed by three researchers (AG, MA, and BS) by reading transcripts, creating codes, sub-codes, and categories, highlighting multiple meanings and key themes in the data and holding regular peer debriefings or meetings to refine codes and meanings (AG, MA, ST, and BS) throughout the analytic stage [26]. The data were thematically analyzed [18] using NVivo 12 analytic software. Quotations included in this article are used to illustrate our results and deepen the understanding of the research questions. Quotations are also woven into the narration of our findings to include many perspectives and add nuance to the interpretation of the data [27]. All participants are anonymized throughout this article to ensure confidentiality. Ethics approval was obtained from the University of Alberta Ethics Board (Pro00115777).

## 3. Results

Table 1 shows the demographic characteristics of the participants included in the study.

Participants were of diverse Black ethnicity who had lived in Canada for an average of 18 years. The majority were single, of Christian faith, university educated, and of Black African heritage.

The results from the study showed a range of responses that highlight the effect of vaccine mandates on COVID-19 vaccine acceptance or refusal. Two major themes arose: acceptance of the COVID-19 vaccine in the context of governmentality and resistance to vaccine mandates driven by oppression, mistrust, and religion. Importantly, participant narratives did not fall into binary approaches to the vaccine. Instead, they demonstrated ways members of this population grapple with fear, hope, and resistance in choosing how to engage with the vaccine. These themes and subthemes highlight various forms of hesitancy concealed within the acceptance of and resistance to vaccine mandates, as shown in Table 2.

### 3.1. Acceptance in the Context of Governmentality

The majority of participants indicated the productive nature of vaccine mandates and situated their usefulness as a tool for reducing the spread of the virus within the historical use of vaccine mandates as a public health promotion strategy. For a participant from Ontario, part of this strategy involved the use of yellow cards or, in the case of the COVID-19 pandemic, vaccine certificates that allowed for the monitoring and surveillance of citizens for the greater good. The majority of these participants framed the benefits of vaccine mandates within already established knowledge of the importance of vaccines, their historical use of vaccines, and vaccine mandates particularly in relation to international travel:

“You know, there is a lot of things we’re no longer dying of and those were vaccine mandates. And we had vaccine mandates, we have always had them first of all. Like think about school, remember those yellow cards—I don’t know how long you have been in Canada, but 30 years ago there were these little yellow cards, and you had all your vaccine history and you walked around with the card. It was like this big and had all your vaccines on it. We had that for years. So, we have had mandates for at least the last 50 years, this is not different…You try to travel someplace; don’t you have to take vaccines if you are going to certain countries?”[Participant 31].

While vaccine mandates can influence the decision-making process by helping individuals “make up their minds faster” [Participant 31], individual protection and community protection including protecting friends and family were key to facilitating vaccine acceptance. For a minority of the participants, acceptance of the vaccine is not straightforward, and the ways in which people, institutions, and structures wield power over people need to be considered. Here, power is not simply power over but also the many ways in which factors, such as employment, affect individual agency on vaccine intentions. “What mandated me taking the vaccine was the job I was about resuming” [Participant 22]. Uptake of the vaccine does not mean the absence of vaccine hesitancy for some participants and highlights the subtle power and nuance of vaccine mandates in vaccine intentions.

Vaccine acceptance also needs to be considered through the lens of obedience. For many participants, obedience is a response to social problems caused by systemic racism and manifests through biopower and governmentality [20]. Biopower thus works in tandem with governmentality to reproduce ways Black people exist that are normalized as their mode of organization in society as highlighted by a participant of West African descent based in Alberta—‘I think the majority of the Black population obey the rules, by respecting mandates and using masks’ [Participant 17]. ‘Obeying the rules’ shows how Black people around the world are conditioned to survive in situations that either diminish their humanity or force them into a mindset of submitting to government authority as a survival tool to exist as a Black person in an increasingly anti-Black world [15,20,28]. Indeed, for participant 17:

“They don’t like it, but they obey it, because of just the nature that they don’t want to—because the majority that I say, you know what, I don’t want to do something. I’m a Black person; if I do something it’s going to be taken in a whole different way.”

Considering the quote above, acceptance of the vaccine can be seen as a survival tool because the repercussions of ‘disobedience’, for example, job loss or unemployment, intersect with race, and actions must be considered within the historical and social context of Black people in Canada. ‘Taken in a whole different way’ underscores the way other Black people (migrants of various Black heritage) experience Blackness in Canada and reinforces the need to conform, to be controlled and docile as they navigate spaces that are overwhelmingly White.

“So, a lot of people do it under duress. They don’t feel like they were able to make an informed consent, so in the end when all the government—the provinces were mandating it for work … and travel and you can’t go anywhere, the people were just taking it more like under duress”

A lack of choice leading to the uptake of the COVID-19 vaccine was featured strongly among many participants as highlighted in the quote above. Many expressed feelings of duress, force, and/or violation of their rights to make personal health choices. Vaccine mandates applied subtle pressure on people to increase vaccination rates. The reward for vaccination was access to work, travel, surgery, entertainment, and other aspects of social and economic life.

### 3.2. Resistance to Vaccine Mandates Driven by Oppression, Mistrust, and Religion

While biopower through vaccine mandates works productively by reducing mortality and morbidity rates [29], and, in the case of COVID-19, taking the vaccine to reduce the prevalence of the disease, it can also be conceived as oppressive, particularly in a situation where society rewards vaccination with access to work, travel, etc. A majority of the participants alluded to the divisive nature of vaccine mandates and the disciplining of unvaccinated people by denying them access to employment, travel, health care (surgeries), and social activities:

“I was just fortunate enough to actually find a job within a non-profit organization that doesn’t require proof of vaccination, but they still require testing twice a week…. But where I’m coming from is, I made the choice [to not get vaccinated], but a lot of people didn’t have the choice, you know what I mean? I know people who literally are on the verge—about to lose their jobs, can’t feed their kids and are the breadwinner for an entire family”[Participant 15].

Resistance, in some cases, cost some individuals their jobs or led to citywide protests against vaccine mandates. In both scenarios, resistance occurred within varying levels of privilege. The choice to remain unvaccinated and seek alternative sources of employment was possible for many people who were unmarried without children or additional responsibilities or who had less fear of remaining unemployed for longer periods. The distinction between vaccinated and unvaccinated people bred discrimination that may have trickled down into the household where couples and other family members do not share the same views on vaccination: “it’s hard to see people discriminate like that against people and classify people as vaccinated and unvaccinated” [Participant 16]; and “she couldn’t participate in events with her partner. So, it was creating separation in their—he got tickets to go to a baseball game in the summertime, she couldn’t go” [Participant 10]. Resistance can also be subtle. For example, many participants expressed the need to wait and see the effect of the vaccine mandates and the vaccine: “Initially I wanted to see how it played out. I wanted to see more long-term effects” [Participant 15].

A majority of participants expressed varying levels of mistrust in the government. Government mistrust operates in people at the level of COVID-19 vaccine reluctance/ hesitancy and anti-vaccination sentiments. Participants with anti-vaccination sentiments were not generally against vaccination and framed their resistance within intersections of narratives centered around an overrepresentation of Black people in medical experiments, the speed of vaccine production, and ‘good citizenship,’—where collective responsibility outweighs individual concern:

“But then for those who aren’t vaccinated, they’re very, very, very adamant in their views and their beliefs. And a lot of that stems from just mistrust of the government and these historical atrocities that have been committed against Black people from vaccination experiments and whatnot”[Participant 28].

In many cases, participants described how mistrust in the healthcare system and knowledge of medical experimentation on Black people and other people of color created a climate of concern and ‘suspicion’ for many Black people in Canada. They explained how vaccine mandates further reinforced these beliefs, particularly considering socio-economic factors that describe the overwhelming presence of Black people in frontline and essential services in Canada. The country of birth for Black immigrants in Canada shapes their mistrust in the government and can be seen as a symptom of a mistrust in the government of their home country and the insecurities attached to governmental control as illustrated in the quote below. This, in addition to a mistrust in the healthcare system and knowledge of medical experimentation on Black people and other people of color, creates a climate of concern and ‘suspicion’ for many participants and, indeed, Black people in Canada.

“I’m always very careful when it comes to using some of this. Especially when they’ve not been tested and proven. So, some people, especially we Africans, because we also need to understand that we Africans do tend to have a bias sometimes in terms of they want to use us as specimens. Those things that—yeah, so we don’t really believe the health system. Probably because we’re coming from a culture where even the politicians are not to be trusted”[Participant 21].

Participants also explained that religion can play a role in vaccine hesitancy. Religious institutions carry influence in many Black communities. Conspiracy theories founded in Christian religious beliefs influenced the way many Black people responded to the pandemic and the vaccine: “They believe there’s going to be anti-Christ, they believe there’s going to be 666, the mark of the beast, and their prophets have said this is how it’s going to happen” [Participant 13]. To some, vaccine mandates encourage the acceptance of the COVID-19 vaccine; however, on the other hand, government interference in individual health choices reinforces narratives within the Christian faith on the coming of the anti-Christ. This highlights a relationship between religion, the government, and science and how they individually and collectively wield power over people.

“So because me, I’m a Christian, right, so when this COVID came up I was looking to the news you know, reading the Bible and also comparing it with what was said, you know, when the time would come you know, so I was very, very skeptical because I was thinking maybe this vaccine is—maybe the 666 we are talking about, right, so I didn’t want to take it. In fact, I told my wife we are not going to take this vaccine. But when our employer made it compulsory, I mean we had no choice; we had to go and be fully vaccinated, so that was my fear. After the vaccination actually nothing happened”[Participant 26].

However, economic pressure was a stronger force for many people to accept the COVID-19 vaccine even when foregrounded against fear and religious beliefs, as illustrated above by a male participant from British Columbia. Oppression works therefore at various levels to influence vaccine intentions, publicly through religion, government, and employment forces.

## 4. Discussion

This article reports qualitative findings from semi-structured interviews with Black people living in Canada who expressed varied responses to vaccine mandates in Canada. We showed that vaccine mandates have reproduced power relations between the government and the self, with the government using political pressure to reduce the spread of the virus and protect the lives of people. As a public health measure, vaccine mandates have a well-established record of controlling the spread of diseases [10]. Within the context of COVID-19, however, the results show a spectrum of responses to vaccine mandates based on their personal experiences and situations including historical sensibilities to past biopolitics and religion. For Black people in this study, vaccine mandates notably interacted with employment, race, and religion. Together, they contributed to both vaccine hesitancy and discourse on the unintended consequences of vaccine mandates on vaccine-preventable diseases. These unintended consequences include some social and political resistance due to human rights violations, stigma against anti-vaxers, and the negative impact of policies in the areas of employment, education, and social life [19].

The results in this article point at a binary response to vaccine intentions—acceptance and resistance—but show variation within this dichotomy and expand our understanding of vaccine acceptance and vaccine hesitancy. Vaccine acceptance is nuanced by knowledge, referred to here in this article as how scientific knowledge on the importance of vaccines promotes acceptance of the COVID-19 vaccine. Results show that individual or historical knowledge of vaccine mandates and disease surveillance fosters trust in vaccine science, and that knowledge reproduced over time, in conjunction with the productive power of vaccine mandates, worked together to foster acceptance and confidence in voluntary and mandated vaccinations [30]. A very important distinction is made within vaccine acceptance—acceptance of the vaccine concealed people’s vaccine intentions, with the mandates serving as stimulants to what can be observed as positive vaccine behaviors. This distinction highlights some of the ways in which the state/government exercises power over its citizens on one hand and citizens operating outside of state/government policies on the other. Acceptance of vaccine mandates presenting as obedience or lack of choice, particularly in the area of employment, is consistent with research that shows the impact of social determinants of health in health-seeking behavior of Black people. Vaccine acceptance rooted in obedience suggests a more problematic notion of acceptance and draws on some of the ingrained systematic forms of oppression experienced by Black people. The previously stated quote “I’m a Black person; if I do something it’s going to be taken in a whole different way” [Participant 17] summarizes the way many Black people experience racial discrimination in their everyday lives. From the courts and justice system to employment, race is one of the social determinants of health that condition Black people’s decision-making processes [31].

Vaccine mandates were implemented in ways that revealed the limitations and exclusionary nature of COVID-19 policies and revealed the complex inequities present in the lives of Black Canadians. For example, Laurencin (2021) reports on the intersection of race, mass incarceration, poverty, and poor access to health care and how these factors contributed to the increase in vaccine hesitancy among Black people [32]. Our results also show a heightened awareness of Blackness, and the disciplining of Black bodies through oppression—enforced through employment or travel, obedience to, or compliance with government authority. Vaccine acceptance expressed as a lack of choice or obedience mirrors obedience in Foucault’s docile body but differs in its sense of urgency. Duress or coercion, as narrated in the data, is present as the threat of loss of employment or inability to travel for various reasons if people choose to remain unvaccinated. The choice to accept vaccination over loss of livelihood points to multiple forms of oppression experienced by visible minorities. For many Black Canadians, the economic effects can be crippling. The acceptance of vaccine mandates drew on the vulnerabilities that are not unique to Black people but have more implications as visible minorities experience more vulnerabilities to COVID-19 than white Canadians due to their prominence in frontline or service sector jobs [33] and plays a key role in participants resistance to vaccine mandates.

Resistance to vaccine mandates and the governmental and systemic power within these policies was identified as a key finding in our study. Resistance to vaccine mandates adds nuance to the ways in which many Black people navigate systems and policies that fail to consider differences across many social categories. While vaccine mandates may have fostered stigma and discrimination [19], our findings show the choice to remain unvaccinated was possible for many at the intersection of class. Here, the fear attached to the loss of livelihood was not a factor, particularly for participants who were non-essential workers or did not have family obligations compared to those who accepted the vaccine to keep their jobs. Research elsewhere in Canada supports this notion, as data showed essential non-healthcare workers and those who identify as non-White, South Asian, or of Indigenous ancestry were less likely to intend to be vaccinated against COVID-19 [33].

Two main factors were employed to contextualize resistance: mistrust (government and healthcare systems) and religion. Our results show mistrust spans historical racism in healthcare including medical experimentations and government mistrust. This echoes some of the issues raised in Charles’s work on HPV vaccination on Black women in Barbados highlighting the agency to resist mandates as intuition based on real or perceived colonial and post-colonial trauma with science and healthcare. Kricorian and Turner (2021) report higher levels of mistrust in government due to medical mistrust among Black Americans [15]. Elsewhere, Keshet and Popper-Giveon (2022) argue that mistrust in the government creates hesitancy through individual assessment of the cost and benefit of taking the vaccine [30]. This assessment plays a key role in determining vaccine acceptance with or without external factors such as religion or mistrust of the government. Religion, including its associated conspiracy or apocalyptic theories, comes together with discrimination against and moral condemnation of the unvaccinated, right-wing fundamentalism, and other conspiracy theories that lend credibility to each other as a source of vaccine hesitancy depending on the social realities of the individual affected [16,19]. However, this study shows the intersection between religious conspiracy theories and employment becomes evident when people choose vaccination over the consequences of being unemployed. While some employers accept a weekly negative COVID-19 test in lieu of fully vaccinated status, employer-imposed vaccine mandates highlight industry/sector and political interference with individual agency and human rights. This, in turn, can reinforce mistrust in government and vaccine hesitancy. Public education, community engagement, and communication with at-risk populations can be improved in order to reduce vaccine hesitancy [34]. Future research should explore the effectual nature of vaccinations including COVID-19 to address lingering post-pandemic anti-vax sentiments and create robust inclusive strategies to inform future response to public health emergencies. This study acknowledges the high level of education among participants compared to the national average for the target population. As such, we encourage future work on vaccine hesitancy including COVID-19 vaccines to account for voices that are more representative of Black people in Canada.

## 5. Conclusions

This article reported results from research on the impact of vaccine mandates on COVID-19 vaccine hesitancy by illustrating how people’s bodies are sites for exercising power and negotiating agency. By describing the range of responses to mandatory COVID-19 vaccination policies among Black people in Canada, this article highlighted the ways in which vaccine mandates influence individual agency and showed the limitations of viewing the uptake of the COVID-19 vaccine wholly as vaccine acceptance. While trust/mistrust in science and the healthcare system can enable/hinder vaccination intentions, using employment to reward or punish individuals has far-reaching consequences for people’s lives.

### Contributions to Knowledge

What does this study add to existing knowledge?

This study contributes to knowledge of COVID-19 vaccine hesitancy among Black people by exploring vaccine acceptance and refusal through lenses of biopower and governmentality.Many of the studies on vaccine hesitancy among people of color are based in the United States. This study adds to the research on vaccine hesitancy among Black people by focusing on the experiences of Black Canadians.

What are the key implications for public health interventions, practice, or policy?

Policy makers need to understand how public health interventions connect to systemic oppression and the ways historical regimes of violence may be reproduced through health mandates.

Policy makers can consider non-discriminatory public health policies and monitor how these policies are implemented over time and across multiple sectors.

## Figures and Tables

**Table 1 ijerph-20-07119-t001:** Demographic characteristics of the participants.

Characteristics		Count (%)
Sex	Men	15 (42%)
	Women	21 (58%)
Marital status	Married	15 (42%)
	Single	16 (44%)
	Common law	1 (3%)
	In a relationship	3 (8%)
	Separated	1 (3%)
Religion	Christian	29 (81%)
	Muslim	1 (3%)
	Atheist	1 (3%)
	Prefer not to say	3 (8%)
	Spiritual but not affiliated	2 (6%)
Average # of years living in Canada		18
Immigration status	Permanent resident	6 (17%)
	Canadian citizen	20 (56%)
	Temporary resident	10 (28%)
Ethnicity	Black African	27 (75%)
	Black American	1 (3%)
	Black Caribbean	8 (22%)
Province	Alberta	13 (36%)
	British Columbia	4 (11%)
	Nova Scotia	3 (8%)
	Ontario	11 (31%)
	Saskatchewan	2 (6%)
	Manitoba	3 (8%)
Education	College	2 (6%)
	High school	2 (6%)
	Masters	1 (3%)
	University (Bachelor’s)	30 (83%)
	Vocational school	1 (3%)
Annual income	<$60,000	17 (47%)
	>$60,000	13 (36%)
	N/A	2 (6%)
	Prefer not to say	4 (11%)

**Table 2 ijerph-20-07119-t002:** Illustrative quotes for each theme and subthemes for various forms of hesitancy concealed within acceptance of and resistance to vaccine mandates.

Acceptance—personal/individual and community protection and return to normalcy	“It not only saves your life, but it protects the community at large” [Participant 1]. “I don’t have any concern with the COVID-19 mandate, and I think the government they will say, take the vaccine or keep testing, you have to test yourself, go in, test yourself, come in. Yeah if people don’t want to do the testing, continuous testing then maybe they are better off doing the, taking the vaccine. I don’t have any concern, I think they’re just protecting the global population. Is it, all be probably good. Yeah, no and the individual do it now, and the government is doing their right, and they’re there to protect the public at large, and not just one person. So I think mandating it for people to do the right thing and take the vaccine, get yourself tested.” [Participant 12].
Acceptance—obedience	“the black community, again who obey rules, laws, always did not complain about a mandate, because they just—and you see that with maybe older folks, who are used to immigrants, older immigrants who knew probably in my country, if they see this, is this is what you obey, these are rules. They look at it and say, “Now, this is what the government wants, I’m going to just do what the government wants.” [Participant 17]. “ I think the vaccine mandates definitely push a lot of people to get vaccinated, but I wouldn’t say that all the time it was willingly. I think for my brother for example. He also lives in Canada and him and his wife, they weren’t against the vaccine, they just were not really in a rush to get it. And for them, they were just like they want to wait a little bit before they get, but then once the mandate started coming up then they were just like, “OK, I guess we don’t really have a choice here. Let’s go.” [Participant 2].
Acceptance—employment	“Because also, as work, it was mandated first. So if I was going to give my permit to the government of ……., I needed to take the vaccine. And so, I had to do that. If it was not mandated that—I probably could still be thinking about it at this time, as the—and the reason is also not farfetched from the fact that I’m not sure.” [Participant 21].
Resistance to vaccine mandates (*n* =5)	
Resistance—Employment	“I, myself, have experienced discrimination and been villainized by not only people I know in my life, but also through my work and my employer, based on my decision to not partake in the vaccine status and passport and all that.” [Participant 15]. “Yes. Like for me, I’ve chosen to wait and not get vaccinated right away. But I also have other tools that I’m using. I’m working from home fortunately. I have a social bubble that I interact with. I’m masking. I would be willing to submit to tests. Give people the effective knowledge that they can make the right decision.” [Participant 10].
Resistance—Religion	“I have friends up to now, they haven’t taken the COVID, because they are men of God or prophets said that it is from the devil. The vaccine is 666. It will take your soul away.” [Participant 30]. “They believe there’s going to be anti-Christ, they believe there’s going to be 666, the mark of the beast, and their prophets have said this is how it’s going to happen. That this COVID-19 is how it’s going to happen and if you bow down to the COVID-19 mandatory vaccine that is how you’re going to bow down when the 666, the mark of the beast, the anti-Christ is going to come. So this is how it’s being propagated in, especially in the Pentecostal and religious circle. And it has made so many people to see COVID-19 as gradually going to be like the mark of the beast, the 666 and the rest.” [Participant 13].
Resistance—government mistrust	“in terms of not necessarily speaking to mistrust in the government, but in terms of the vaccinations specifically with COVID-19, I think that the method at which the government went about enforcing these policies just rubbed a lot of people the wrong way—between the propaganda, the censorship, the bribery and then the coercion—it’s just not necessarily painting a good picture, in the sense of people aren’t necessarily willingly taking the vaccine because of the benefit they think it will be for themselves personally, but rather of the narrative that has been spun about doing your part for your community” [Participant 15]. “…not only are individuals who don’t participate in the vaccine passport—not only are they discredited, and their character called into question because this is now an issue of morality.” [Participant 2].
Resistance—mistrust in healthcare system/service providers	“And then just going back to the Tuskegee experiment as well, just going into Black communities and just saying you know, this will work for you. Here are the effects. So yeah, that has influenced me. But more so I would say it is the science. I just don’t think enough has been told to us about—is the number 1 influence about this vaccination. I just have concerns about what is in it. Is this a true vaccination and reporting on the adverse effects. So again, when something is being pushed, or I hear come to this vaccine clinic and they’re handing out $50 gift cards, that to me is very suspect. Or they’re having bouncy castles and clowns and smiley faces for the kids when they are. Or superhero themes. Or giving out, you know, restaurant vouchers. Grocery vouchers” [Participant 10].

## Data Availability

The data presented in this study are available on request from the corresponding author. The data are not publicly available due to ethical review policies at the University of Alberta.

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
