# Peer review of "COVID-19 Vaccine Mandates and Vaccine Hesitancy among Black People in Canada"

_ijerph, 2023, doi:10.3390/ijerph20237119_

Round 1

Reviewer 1 Report

Comments and Suggestions for Authors

The manuscript by Aisha Giwa et al analyzed COVID-19 vaccine mandates and vaccine hesitancy by semi-structured interviews with 36 Black people. There are some issues that need to be addressed before this paper can be considered suitable for publication on IJERPH.

Major concerns:

1.      Only 36 people were enrolled in semi-structured interviews.

2.      Besides one table of demographic characteristics of the participants, there were no tables or figures regarding the research conclusion.

3.      The interview should also enroll White people or Asian people as control.

Comments on the Quality of English Language

The quality of English writing is acceptable and readable.

Author Response

  1. Only 36 people were enrolled in semi-structured interviews.

Response: 36 people were enrolled in the study for the following reasons:

  1. Response rate (36 participants responded to our invitation to be interviewed on their experiences and perspectives on COVID-19, COVID-19 vaccines and ways to improve vaccine confidence and uptake among Black people living in Canada)
  2. Informational redundancy (sampling can be terminated when no new information is elicited by sampling more participants[1])
  3. Adequate sample size: Researchers have proposed 30[2] or a range between 20 and 30 interviews [3] as an approximate or working number of interviews at which one could expect to reach theoretical saturation when using a semi-structured interview approach.
  4. Besides one table of demographic characteristics of the participants, there were no tables or figures regarding the research conclusion.

Response: A table showing the overarching individual-level responses to COVID-19 vaccine mandates has been included in the results section.

  1. The interview should also enroll White people or Asian people as control.

Response: Black Canadians have the highest rate of COVID-19 infection and also the lowest rate of COVID-19 vaccination in Canada. Improving the COVID-19 vaccination rate among Black Canadians is of great interest to policymakers and stakeholders. Our project seeks to address knowledge gaps related to COVID-19 vaccine confidence and uptake among Black Canadians and inform effective strategies to improve COVID-19 vaccine confidence and, in turn, uptake among Black Canadians. Other populations including White or Asian people cannot be included in this study.

[1] Lincoln YS and Guba EG. Naturalistic inquiry. London: Sage; 1985.

[2] Morse JM. Data were saturated. Qual Health Res. 2015;25(5):587–8.

[3] J.W. Creswell (1998) Qualitative Inquiry and Research Design: Choosing Among Five Traditions Sage Thousand Oaks, CA

Reviewer 2 Report

Comments and Suggestions for Authors

The authors present a qualitative study on the reasons behind vaccine hesitancy among Black people in Canada. The study is based on 36 interviews. Demographic characteristics of the participants show a good mix of socioeconomic variables on average. The discussion is detailed and makes a good job at tying together the results of the study with contemporary literature. I believe the topic is very important and relevant for future vaccination campaigns. I have the following questions:

1. Table 1 shows that most of the participants (83%) have a university level education. Is that common in the target demographic? What is the explanation behind this?

2. Maybe because I have a quantitative background, but I would like to see more numbers in the results part. The quoted sentences provide a good summary of the beliefs of specific individuals, but how common are these beliefs among the rest of the group? Is there any way you could show this?

3. In section 2 you mention a coding framework and thematic analysis. Maybe you could present details of this framework? Or the thematic analysis? It would be useful for the reader.

Author Response

  1. Table 1 shows that most of the participants (83%) have a university level education. Is that common in the target demographic? What is the explanation behind this?

Response: the educational attainment of the participants aligns with reports from Statistics Canada[1] that show that Black people have similar educational attainment with non-Indigenous non-racialized population but are less likely to find a job that pays as well. The link to the StatsCan report can be found here.

  1. Maybe because I have a quantitative background, but I would like to see more numbers in the results part. The quoted sentences provide a good summary of the beliefs of specific individuals, but how common are these beliefs among the rest of the group? Is there any way you could show this?

Response: A table showing the overarching individual-level responses to COVID-19 vaccine mandates has been included in the results section.

  1. In section 2 you mention a coding framework and thematic analysis. Maybe you could present details of this framework? Or the thematic analysis? It would be useful for the reader.

Response: a brief description of the coding framework used in thematic analysis is included in the methods section. 

[1] https://www150.statcan.gc.ca/n1/pub/75-006-x/2023001/article/00009-eng.htm

Reviewer 3 Report

Comments and Suggestions for Authors

This was an interesting study.  However, although the authors said that the majority of respondents supported vaccine mandates (p. 4), it emphasized distrust and resistance.  Even in the section that noted widespread support, most of the space was devoted to arguing this was the result of coercion--e. g., "'Obeying the rules' show how  Black people around the world are conditioned to survive in situations that . . . diminish their humanity or force them into a mindset...."  So I think there should be more discussion of the positive motives for support of vaccines or vaccine mandates as well as the motives for opposition.  Also, although with a small, non-representative sample you can't expect precise estimates, it would be helpful to have some sense of how views are distributed--that is, roughly how many are basically favorable, basically unfavorable, ambivalent, ....  There should also be some discussion of political divisions over the issue and how they affected the respondents. 

One other point that deserves attention--the sample was more highly educated than the general public.  How might that have affected the findings?  It would also be useful to have some information on the occupational distribution. 

Author Response

The results showed that support for the mandates did not mean that participants did not experience vaccine hesitancy. We attempted to show the complexity of viewing acceptance at face value, the same with resisting vaccine mandates. We have made some edits to the discussion section with the intent of clearly discussing the data we presented. Political divisions over the issue did not appear significantly in the data so we didn’t include that in the results.

The study did not collect details of the participant’s occupation. However, the educational attainment of the participants aligns with reports from Statistics Canada[1] that show that Black people have similar educational attainment to non-Indigenous non-racialized populations but are less likely to find a job that pays as well. There is some relationship between education particularly those that have a science-related degree and understanding the severity of COVID-19.

[1] https://www150.statcan.gc.ca/n1/pub/75-006-x/2023001/article/00009-eng.htm

Reviewer 4 Report

Comments and Suggestions for Authors

Influencing the sometimes weak willingness to vaccinate has become a key issue in the fight against COVID-19. Conspiracy theories circulating on the Internet and insufficient health literacy were a serious obstacle to taking effective jabs.

In this manuscript, the possibility of reducing vaccine hesitancy with the help of a vaccine mandate among black Canadians was investigated using semi-structured interviews.

The hypothesis of the draft was that any coercion would only increase the insecurity and resistance of the population. The interviews seemed to confirm this, although in terms of the methods, it would have been appropriate to present the questions and to illustrate and support the evaluation with more than just anecdotal quotes. More data and analysis are needed regarding the income situation, whether skin color and/or only the degree of poverty or lack of education play a role in aversion to vaccines. Examination of control groups may also be justified.

I believe that in the discussion, it would be possible to take a clearer position in favor of the fact that the improvement of the willingness to vaccinate should not be achieved with means that cause stigmatization, or that are punitive or perceived as such, but with motivating, informative, and educational measures. The paper therefore needs to be supplemented. The application of more precise wording should also be considered.

Author Response

The introduction section, discussion and methodology have been updated. A table showing the overarching individual-level responses to COVID-19 vaccine mandates has been included in the results section to further illustrate the results from our study. The authors agree that more data and analysis are needed regarding the income situation, whether skin color and/or only the degree of poverty or lack of education play a role in aversion to vaccines. We suggest that further quantitative research can be conducted to unpack the relationship between income, education and race on peoples vaccine intentions.

Round 2

Reviewer 1 Report

Comments and Suggestions for Authors

Authors have answered my concerns.

Author Response

No response is required. 

Reviewer 2 Report

Comments and Suggestions for Authors

I see the manuscript has been significantly improved. I accept the authors' responses to my last two questions.

I still find it notable that 83% of the respondents had a university level education. According to the document the authors cite "When viewed at a high level, the share of the Canadian-born Black population aged 25 to 54 who have a bachelor’s degree or higher (29%) ... The African-origin Black population were the most likely to have a bachelor’s degree or higher, at 46%, compared with 27% among the Caribbean-origin Black population and 16% among the Canadian-origin Black population (Chart 1)."

This might simply be a misunderstanding. The authors have college-high school-masters-university-vocational school as education categories. So what does university mean in this setting? Bachelors? This should be clarified in the manuscript. Even so, it seems like the respondents were better educated than the target demographic, and this should be noted as a limitation of the study.

Author Response

Thank you for the feedback. I have added a few sentences on the level of education as a limitation of the study

Reviewer 3 Report

Comments and Suggestions for Authors

I think this revision adequately addresses the comments from the previous round.  A few minor points:

1.  The first sentence should say something stronger than "argued"--maybe "shown" or "demonstrated".

2.  The second sentence defines vaccine mandates, but doesn't finish the thought (they've been proposed as a way to increase vaccination rates?  they've been controversial?)

3.  The lack of a clear political division they mention in their reply to me also deserves mention in the text.

4.  On education, the table classified 80% of this sample as "university"; the source they cite shows 20%-60% of different subgroups as having a BA.  So it still appears that this sample is highly educated relative to the relevant population--that should be mentioned as a limitation.  

5.  It occurs to me that there should also be some mention of how many of the respondents were born in Canada and how many were immigrants. 

Author Response

Thank you for the feedback. I have responded to issues raised in #1-4 and included a few sentences on the limitations of the study. I will not be able to respond to the last comment because the study did not collect information on the birth history of the participants - whether or not they were born in Canada. It is possible to infer from the data in the table that many of the participants were born outside of Canada (n-16). However, there is no way to tell for those who identified as Canadian citizens. 

Reviewer 4 Report

Comments and Suggestions for Authors

The manuscript has significantly improved as the authors took into account the reviewer's opinion and advice and revised the draft accordingly. 

Author Response

No response is required.